# Molecular Mechanisms of Phosphate Sensing, Transport and Signalling in *Streptomyces* and Related Actinobacteria

**DOI:** 10.3390/ijms22031129

**Published:** 2021-01-23

**Authors:** Juan Francisco Martín, Paloma Liras

**Affiliations:** Department of Molecular Biology, Microbiology Section, University of León, 24071 León, Spain; paloma.liras@unileon.es

**Keywords:** phosphate, transport systems, PstS, PitH, *Streptomyces*, sugar phosphates, glycerol-3-phosphate, phosphonates

## Abstract

Phosphorous, in the form of phosphate, is a key element in the nutrition of all living beings. In nature, it is present in the form of phosphate salts, organophosphates, and phosphonates. Bacteria transport inorganic phosphate by the high affinity phosphate transport system PstSCAB, and the low affinity PitH transporters. The PstSCAB system consists of four components. PstS is the phosphate binding protein and discriminates between arsenate and phosphate. In the *Streptomyces* species, the PstS protein, attached to the outer side of the cell membrane, is glycosylated and released as a soluble protein that lacks its phosphate binding ability. Transport of phosphate by the PstSCAB system is drastically regulated by the inorganic phosphate concentration and mediated by binding of phosphorylated PhoP to the promoter of the PstSCAB operon. In *Mycobacterium smegmatis*, an additional high affinity transport system, PhnCDE, is also under PhoP regulation. Additionally, *Streptomyces* have a duplicated low affinity phosphate transport system encoded by the *pitH1–pitH2* genes. In this system phosphate is transported as a metal-phosphate complex in simport with protons. Expression of *pitH2*, but not that of *pitH1* in *Streptomyces coelicolor*, is regulated by PhoP. Interestingly, in many *Streptomyces* species, three gene clusters *pitH1–pstSCAB–ppk* (for a polyphosphate kinase), are linked in a supercluster formed by nine genes related to phosphate metabolism. Glycerol-3-phosphate may be transported by the actinobacteria *Corynebacterium glutamicum* that contains a *ugp* gene cluster for glycerol-3-P uptake, but the *ugp* cluster is not present in *Streptomyces* genomes. Sugar phosphates and nucleotides are used as phosphate source by the *Streptomyces* species, but there is no evidence of the *uhp* gene involved in the transport of sugar phosphates. Sugar phosphates and nucleotides are dephosphorylated by extracellular phosphatases and nucleotidases. An isolated *uhpT* gene for a hexose phosphate antiporter is present in several pathogenic corynebacteria, such as *Corynebacterium diphtheriae*, but not in non-pathogenic ones. Phosphonates are molecules that contains phosphate linked covalently to a carbon atom through a very stable C–P bond. Their utilization requires the *phnCDE* genes for phosphonates/phosphate transport and genes for degradation, including those for the subunits of the C–P lyase. Strains of the *Arthrobacter* and *Streptomyces* genera were reported to degrade simple phosphonates, but bioinformatic analysis reveals that whole sets of genes for putative phosphonate degradation are present only in three *Arthrobacter* species and a few *Streptomyces* species. Genes encoding the C–P lyase subunits occur in several *Streptomyces* species associated with plant roots or with mangroves, but not in the laboratory model *Streptomyces* species; however, the *phnCDE* genes that encode phosphonates/phosphate transport systems are frequent in *Streptomyces* species, suggesting that these genes, in the absence of C–P lyase genes, might be used as surrogate phosphate transporters. In summary, *Streptomyces* and related actinobacteria seem to be less versatile in phosphate transport systems than Enterobacteria.

## 1. Introduction

Inorganic phosphate (Pi) is a key nutrient in cell metabolism, since it is constitutive of nucleic acids, phospholipids, teichoic acids, membranes, highly phosphorylated nucleotides, and phosphorylated proteins. It participates in the respiratory chain, and plays an important role in the signalling cascades [1,2]. Phosphate salts are present in a variety of soil and aquatic environment, but its concentration in different habitats is very diverse. Sometimes phosphate is present in insoluble form (so called rocks phosphate) that is difficult to metabolize and requires its solubilization by microorganisms [3,4]. In addition, organophosphates are abundant in some particular habitats as products of decay of plant and animal tissues.

*Streptomyces* are largely soil dwelling organisms, although they are also present in many aquatic environments, including sea waters. *Streptomyces* species and related actinobacteria are prolific producers of a plethora of secondary metabolites with useful biological activities [5,6]. The biosynthesis of these metabolites in *Streptomyces*, filamentous fungi, and plants is controlled by the phosphate concentration in the cells, reflecting the availability of phosphate in the different environments [7]. Phosphate regulates the biosynthesis of secondary metabolites belonging to diverse biosynthetic classes, including polyketides, non-ribosomal peptides, amino acid-derived metabolites, isoprenoids, and complex antibiotics, suggesting that the regulation by phosphate is a widely conserved mechanism that controls: (1) primary metabolism and, therefore, the availability of precursors for the biosynthesis of secondary metabolites, and (2) the expression of the genes encoding these biosynthetic pathways [8]. 

Phosphate limitation causes a well-known nutritional stress that triggers expression of genes encoding secondary metabolite biosynthetic enzymes [9,10]. In response to phosphate limitation, *Streptomyces* species produce extracellular enzymes, including alkaline phosphatases PhoA, PhoC, the Ca^2+^-dependent phospholipase PhoD [11,12], and a eukaryotic-type phosphatase, PhoX [13,14]. A response of *Streptomyces* and other microorganisms to the phosphate availability in the environment is to maintain a reservoir of phosphate in the form of polyphosphate [15,16]. This compound is accumulated in large amounts in *Streptomyces* when grown in phosphate rich media and this allows maintenance of *Streptomyces* metabolism for a prolonged time in phosphate starvation conditions. 

Expression of hundreds of genes is controlled by the availability of phosphate; these genes constitute the Pho regulon [1,17,18,19]. Expression of the Pho regulon in *Streptomyces* is controlled by the two-component system PhoR–PhoP [20,21]. The mechanism controlling the Pho regulon has been reviewed elsewhere [2,22] and is not detailed here.

Since organic and inorganic phosphate are frequently limiting in some habitats, research on the phosphate transport mechanisms is really important. We focus this article on the mechanisms of phosphate transport into the cells in *Streptomyces* and closely related actinobacteria. Most of the available information has been reported in the model actinomycetes *Streptomyces coelicolor* or *Streptomyces lividans*, and in *Corynebacterium glutamicum.* Detailed information on phosphate transport in *Mycobacterium tuberculosis* is not included because it has been extensively discussed [23,24,25,26,27].

In this article we address first the high affinity *pstSCAB* and the low affinity *Pit* phosphate transport systems, and we will discuss later the presence of alternative phosphate transporters.

## 2. High Affinity Phosphate Transport Systems 

Most bacteria, including *Streptomyces* and related actinobacteria, contain high affinity and low affinity phosphate transport systems, which are induced by phosphate limitation, although they may respond to different phosphate thresholds for induction. The wide distribution of both the low affinity and high affinity systems in Gram positive and Gram- negative bacteria indicates that there are physiological advantages in having two different transport systems. The low affinity Pit system requires less energy for phosphate transport since it is energized by a proton-symporter system and is used in high phosphate concentration habitats. The high affinity PstSCAB requires more energy for phosphate transport as it is energized by ATP hydrolysis. This high affinity system is used in habitats or environmental conditions in which phosphate is limiting.

Early evidence of the existence of two phosphate transport systems in *Streptomyces* was provided by the description of two distinct phosphate uptake kinetics in *Streptomyces granaticolor* [28], although the genes encoding these systems were not identified.

### 2.1. The High Affinity PstSCAB System

The high affinity *pstSCAB* was first studied in *Escherichia coli* [29], *S. lividans* [20] and *Bacillus subtilis* [30]. In these bacteria, the system consists of four proteins PstS, PstC, PstA, and PstB, but in *B. subtilis* there are two copies of PstB (PstB1 and PstB2). The system in *Streptomyces* and most other bacteria is organized as an ATP-binding cassette (ABC) that comprises four proteins. Of these, PstS is a phosphate specific binding protein that is anchored in the outer site of the membrane in Gram-positive bacteria, e.g., in *Mycobacterium bovis* or *S. lividans* [25,31,32], whereas in Gram-negative bacteria, it is located in the periplasmic space. PstC and PstA are membrane integral proteins that form a membrane channel and PstB is an ATP hydrolysing protein that energizes the phosphate transport (Figure 1). The complete genome sequence of S. *coelicolor* [33] allowed to identify the *pstSCAB* operon that is conserved in most *Streptomyces* species. Expression of the *pstSCAB* cluster responds drastically to the phosphate concentration in the culture medium and is strictly dependent of the activation by the phosphate regulator PhoP [20,21,34]. Deletion of the *pstS* gene of *S. lividans* or *S. coelicolor* impaired phosphate transport and delayed sporulation in solid medium [34]. In addition to regulation by the PhoP master regulator, formation of the PstS protein was highly increased in a mutant deficient in polyphosphate kinase [34,35,36] which catalyses, in vitro, the reversible polymerization of the phosphate from ATP into polyphosphate. This is most likely due to the up-regulation of *phoR/phoP* observed in the *ppk* mutant strain [15]. The formation of PstS increases significantly in media containing fructose, galactose, or mannose, suggesting that control of the *pstSCAB* operon is regulated by both inorganic phosphate and some carbon sources. 

DNA protein binding studies using the purified PhoP protein of *S. coelicolor* showed a PHO box consisting of two 11 nucleotides direct repeat units (DRu) upstream of the *pstS* gene of *S. coelicolor* that adjust to the consensus PhoP-binding sequence [21,37]. The *pstSCAB* operon of *S. lividans* is expressed as a single transcription unit, although levels of the *pstS* gene transcript were higher than those of other genes of the operon [32]. Most likely this is due to the presence of transcription termination sites between the *pstS* and *pstC* genes. In *S. lividans*, deletion of the region encoding the PHO boxes decreased the expression of the *pstSCAB* operon, but allows a remaining basal level of expression (about 10%) that was controlled by the carbon source [32]. The molecular mechanism of regulation by fructose or other carbon sources has not been fully elucidated. CRP (cAMP receptor protein)-binding sequences have been recently identified in the genome of *Streptomyces roseosporus* [38], and a twelve nucleotide repeated sequence upstream of the *pstS* gene [32] might correspond to a CRP binding sequence.

### 2.2. Sensing the Phosphate Limitation: The Seven Proteins Model of Wanner and the Intracellular Teichoic Acid Intermediate Signal Model

Following the discovery of the two-component system PhoR–PhoB and the *pstSCAB* phosphate transport system in *E. coli*, models were proposed to try to understand which is the sensing mechanism and, particularly, the transduction cascade that responds to different phosphate concentrations in the periplasmic space. Hsieh and Wanner [39] elaborated a refined model in which they proposed that seven proteins interact at the membrane level, serving to detect the phosphate concentration in the broth and then transmit the signal to control expression of the Pho regulon genes. This model proposes that the four proteins, PstSCAB, of the ABC transporter system interact with PhoR–PhoB and also with the phosphate modulator PhoU (Figure 1). According to these authors, depending on the concentration of phosphate in the culture broth, the Pst transporter is either in: (1) an active signalling and transporter state (so called closed state) or, (2) in a state in which it acts as a transporter, but does not transmit signals (open state). The phosphate concentration dependent change between both states produces a rearrangement of the seven proteins complex at the cell membrane, that under phosphate limitation conditions results in autophosphorylation of PhoR, which in turn transfers the phosphate group to PhoB. Finally, the phosphorylated PhoB binds the specific sequences in the promoters of genes of the Pho regulon (Figure 1).

In the *Streptomyces* species, the gene *phoU*, encoding a modulator of the *phoR*–*phoP* expression, is not linked to the *pstSCAB* cluster, as occurs in *E. coli*, but instead is attached to the *phoR*–*phoP* cluster [15,40]. Signalling of the phosphate concentration and its transport by the PstSCAB cassette in *Streptomyces* has received some attention, but is still unknown if the signal interacting with PhoR is internal or extracellular. A second model proposes that, as occurs with *B. subtilis*, the signal is an intermediate in the teichoic acid biosynthesis that inhibits the PhoR autokinase activity and, therefore, decreases the phosphorylation of PhoP and the subsequent activation of the Pho regulon [2,41]. This correlate well with the presence in *S. coelicolor* of a gene cluster for teichoic acid biosynthesis regulated by phosphate similar to that of *B. subtilis* [42]. Under phosphate limiting conditions the phosphorylated teichoic acid intermediate is not accumulated, releasing the inhibition of the PhoR autokinase and the PhoR–PhoP signal transduction cascade proceeds to activate the Pho regulon (Figure 1B).
Figure 1Phosphate transport and signalling according to the seven proteins and the teichoic acid intermediate models. (**A**) The seven proteins model of Hsieh and Wanner [39]. (**Left**) In phosphate replete conditions, the PstS protein binds phosphate and this interaction blocks the autophosphorylation of PhoR and the subsequent phosphorylation of PhoP avoiding the activation of the genes of the Pho regulon. (**Right**). When phosphate is limiting the PstS conformation changes, and triggers the phosphorylation cascade, resulting in PhoP phosphorylation and activation of the Pho regulon. (**B**) Proposed model of regulation of the phosphorylation cascade by an intermediate of the teichoic acid biosynthesis (red circle). Under phosphate replete conditions (left side) the teichoic acid intermediate accumulates and inhibits the autokinase activity of PhoR, blocking the phosphorylation cascade. Under phosphate limitation conditions (right side), the teichoic acid intermediate is not accumulated [41] and, therefore, the phosphorylation cascade proceeds activating expression of the Pho regulon genes (see text for details).
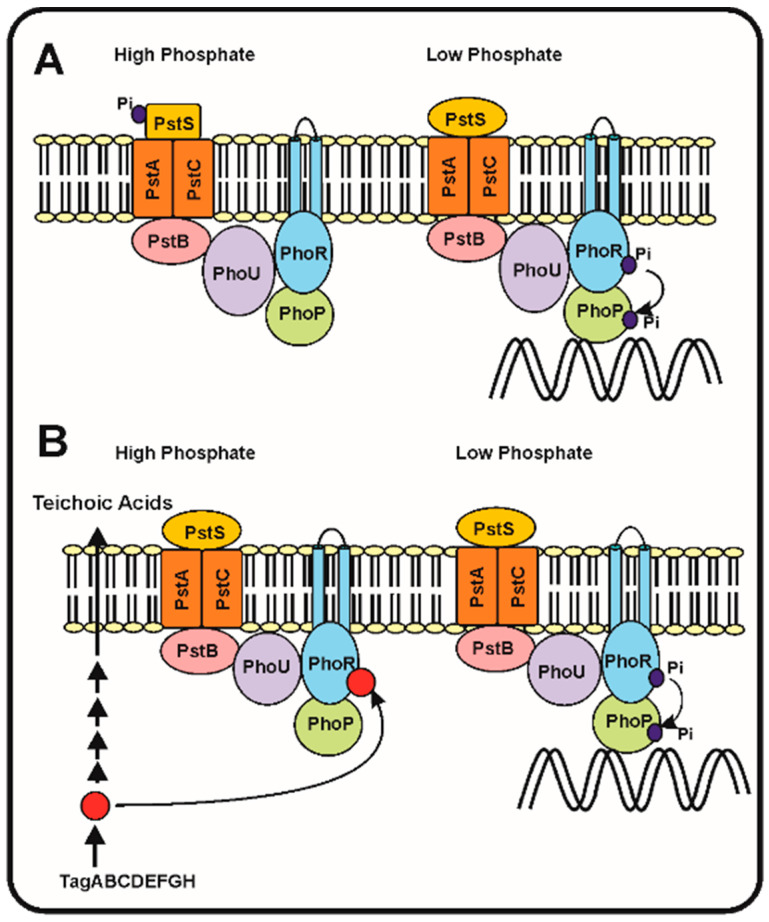

Figure 2Supercluster of genes involved in phosphate uptake and metabolism in *Streptomyces* species. The central region includes the *pap–pitH1* operon (shadowed in pink), the *pstSCAB* cluster (green), the *pptA–nudA* genes (yellow) and the *ppk* gene (blue). In the upper part, the PHO boxes upstream of *pstS* in *S. coelicolor* and *Streptomyces tsukubaensis* are shown indicating the direct repeat units (Dru’s) (boxed) and their individual information values (R_i_). In the lower part updated PHO boxes upstream of the *ppk* gene in *S. coelicolor* and *S. tsukubaensis* are shown. Note that the *pap–pitH1* operon is not regulated by phosphate and does not contain PHO boxes.
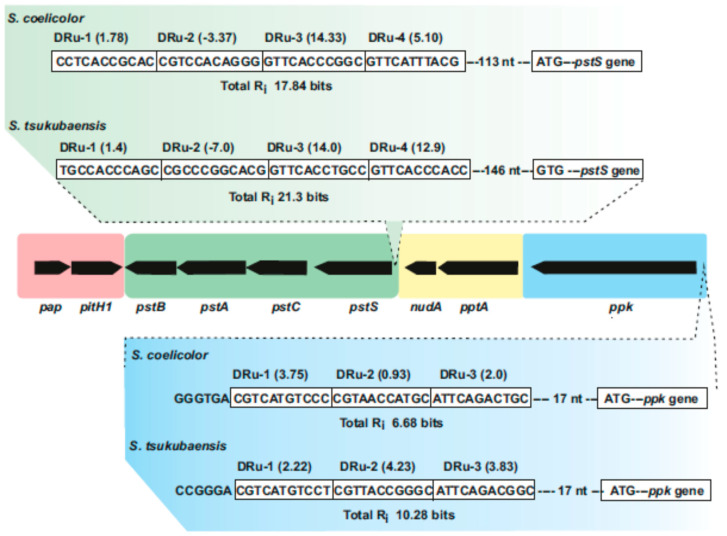



A third hypothesis is also possible: PhoU and the metalloprotein encoded by *mtpA*, a gene adjacent to *phoU*, might be involved in the sensing or resistance to oxidative stress [43,44]. Since PhoU it is known to modulate expression of the *phoR–phoP* operon [15,40], it is possible that oxidative stress might lead to a modified PhoU conformation conferring to PhoU the ability to stimulate the PhoR autophosphorylation activity. Studies on the conformational changes of PhoU are required to support this hypothesis. 

### 2.3. Discrimination between the Phosphate and Arsenate Anions by the Phosphate-Binding Protein PstS

Phosphate (PO_4_^3−^) and arsenate (AsO_4_^3−^) are equivalent anions with similar pK(a) values and charges in their atoms. Arsenate, a toxic anion, is present in some terrestrial and aquatic habitats and, therefore, *Streptomyces* and other bacteria face the problem of discriminating between phosphate and arsenate. In previous studies, we observed in the polyene macrolide antibiotic candicidin producer *Streptomyces griseus*, that arsenate is toxic at concentrations above 10 mM. Expression of the genes for candicidin biosynthesis and candicidin formation are strictly regulated by the phosphate concentration in the cultures [45,46]. Some mutants resistant to arsenate (up to 100 mM) are impaired in the phosphate transport, as determined by radioactive phosphate uptake experiments, whereas others showed normal phosphate uptake [47]. This suggests that somehow there is a competition between phosphate and arsenate, either in the phosphate transport system or perhaps downstream in the signal transducing mechanisms. Several mutants resistant to arsenate overproduced candicidin at phosphate levels (10 mM) that were repressive for candicidin biosynthesis in the parental strain. The molecular mechanism responsible of the candicidin overproduction have not been studied so far. 

The PstS proteins have a great affinity for phosphate with Kd values in the submicromolar range. The affinity of the PstS protein for phosphate is 500 to 700 higher than that for arsenate [48]; even more, the PstS protein of a *Halomonas* strain isolated from a arsenate rich environment has 4500-fold higher affinity for phosphate than for arsenate. Crystal structure of several bacterial PstS proteins [27,49] indicates that they are composed by two globular domains linked together by a flexible hinge. At the interface between the domains, there is a phosphate binding pocket containing a key aspartate residue. Twelve to fourteen hydrogen bonds are formed between the phosphate and the binding pocket, and the aspartate residue establishes an additional low barrier hydrogen bond. The presence of both, high and low hydrogen barrier bonds, confers high selectivity for phosphate. A PstS protein with higher affinity for arsenate has been crystalized from *Clostridium perfringens**;* this protein still forms 14 hydrogen bonds, but lacks the low barrier bond between the phosphate and the aspartate residue [49]. The arsenate anion is 4% larger than the phosphate anion; its accommodation in PstS requires the distortion of this low barrier hydrogen bond allowing its discrimination by distorting this low barrier hydrogen bond. No PstS proteins of *Streptomyces* have been yet crystalized to confirm whether similar mechanism occurs in these bacteria.

### 2.4. Glycosylation and Release of PstS in Streptomyces Species: Which Is the Role of the Released PstS Protein?

The PstS proteins are usually cell membrane-anchored proteins in different bacteria. However, surprisingly it was reported that in some *Streptomyces* species this protein was released to the culture medium.

The PstS protein of *S. coelicolor* is glycosylated [31]. Glycosylation in actinobacteria has been well studied in *M. tuberculosis* and proceeds through O-mannosylation of threonine or serine residues located in a proline rich region of the target protein. Several extracellular antigens are glycosylated in *M. tuberculosis*, including the 45/47 kDa, the 19 kDa and the main 38 kDa antigen protein, identified as PstS1 [23,24]. The carbohydrate moieties include mannose, mannobiose and mannotriose [24]. The glycosylation proceeds first through activation of mannose with polyprenol phosphate by a polyprenol phosphate mannose synthase (Pmm). A second gene involved in the glycosylation process is *pmt1* that encodes a polyprenol mannose transferase.

*S. lividans* contains homologous genes encoding the same enzymes than *M. tuberculosis.* The first evidence of the involvement of these two enzymes in glycosylation in *Streptomyces* came from studies of the glycosylation of two phage receptor proteins in *S. coelicolor* [50,51]. The enzymes encoded by the *pmm* and *pmt1* genes of *S. lividans* were able to glycosylate antigens and extracellular enzymes of *M. tuberculosis* [52,53] confirming the similarity of the glycosylation systems in both actinobacteria. In addition to its role in the synthesis of glycoproteins that serve as receptor of phages in *S. coelicolor*, Wehmeier et al. [31] found that these two enzymes were also involved in the glycosylation of the PstS phosphate binding protein. In *S. coelicolor* glycosylation involves the introduction of three hexose molecules and does not occur in *pmt1* or *pmm* impaired mutants [31]. The two enzymes were functional to glycosylate PstS peptides bound to membranes of *S. coelicolor.* The physiological role of the PstS modification by glycosylation is unclear. Glycosylation of proteins in prokaryotic and eukaryotic cells is connected with protein sorting, protein–protein interactions and membrane localization. 

The PstS protein of *S. lividans* is released from the mycelium to the culture broth, and interestingly the free form is unable to bind phosphate at difference of the cell-anchored PstS protein. In summary, expression of the *pstS* gene in *S. coelicolor* drastically increases in response to phosphate starvation and, therefore, the protein is accumulated in the cell membrane and under adequate conditions is released. In some bacteria, it has been reported that the PstS protein has additional roles different from serving as the phosphate receptor protein. For example, in multi-drug resistant *Pseudomonas aeruginosa* the extracellular PstS forms appendage-like structures that interact with human epithelial intestinal cells disturbing their permeability [54]. It is then intriguing to know if the release of the PstS protein of *S. coelicolor* may have functions resulting in interaction and communication of the producer *Streptomyces* species with other soil microorganisms or with animal tissues in the case of pathogenic *Streptomyces* species, although there is no evidence so far supporting this hypothesis. 

### 2.5. A Second High Affinity Phosphate Transport System Exists in Mycobacterium smegmatis 

The slow growing actinobacteria *M. tuberculosis* contain several high affinity transport systems similar to *pstSCAB* (two to four copies) whereas fast growing mycobacteria contain usually a single copy of the *pstSCAB* cluster. The PstS1, Pst2, and PstS3 phosphate sensor proteins of *M. tuberculosis* have been studied in detail [26] and some of them have been crystalized [27]. The fast growing *Mycobacterium smegmatis* contains two high affinity phosphate transport systems of different types; in addition to the classical *pstSCAB* phosphate transport system, a second transport system named *phnFDCE* (so named because it was initially proposed to be a phosphonate transporter) was characterized [55,56]. The PhnDCE transport system, an ABC-type transporter, consists in three structural proteins: *phnC* encodes an ATPase, *phnD* a phosphate-binding protein and *phnE* encodes a permease. The *phnF* gene, transcribed in opposite orientation encodes a regulator of the GntR family that controls the *phnCDE* cluster [57]. The PhnF protein represses expression of the *phnDCE* cluster by binding to two nucleotide sequences in the bidirectional promoter regions [57]. This system transports Pi with relatively high affinity (Km values of 40 to 90 micromolar), and does not transport phosphonate or phosphite, a salt of phosphorous acid [58]. Expression of the *phnDCE* cluster increased under phosphate limitation conditions. Mutants defective in the phosphate-binding protein PhnD, failed to grow in minimal medium containing at high phosphate concentrations (10 mM) while the parental strain requires only micromolar phosphate concentrations. Complementation of the null *phnD* mutant with the wild type gene resulted in recovery of the growth of this strain at submicromolar phosphate concentrations [55]. In summary, the available evidence indicates that the *M. smegmatis phnFDCE* transport system is a second high affinity transporter for inorganic phosphate at difference of the Phn system in Gram negative bacteria that has been reported to transport both phosphonate and inorganic phosphate [59,60,61,62]. Gene clusters similar to *phnCDE* exist in many *Streptomyces* species, but only occasionally they are linked to the phosphonate transport and utilization clusters (see below).

## 3. Low Affinity Phosphate Transporters

The low affinity inorganic phosphate transport (Pit) systems have been studied only recently in *Streptomyces* species; background knowledge relies in detailed studies performed in *E. coli* or other Gram-negative bacteria.

### 3.1. Transport by the Inorganic Phosphate (Pit) System: Background Information in Gram-Negative Bacteria 

Pit transport systems were initially reported in *E. coli* [63,64,65] and *Sinorhizobium meliloti* [66]. Divalent cations, such as Mg^2+^, Ca^2+^, Co^2+^, or Mn^2+^, are essential for the activity of this transport system in both *E. coli* [67] and *Acinetobacter*
*johnsonii* [68]. The low affinity transporter appears to prefer Zn^2+^ or Mg^2+^, forming a neutral metal-phosphate complex (Me-PO4) [69,70,71]. Using membrane vesicles of *E. coli* Van Veen et al. [68] characterized the kinetics and metal requirements of the Pit system that transports simultaneously equimolecular amounts of phosphate and a divalent metal. This complex is driven by a proton symport mechanism with a stoichiometry of 1:1 (Me-PO4 to proton). In some microorganisms the Pit system also secretes phosphate by combination with heavy metals forming the Me-PO4 complex [72]. This mechanism serves to detoxify heavy metals and uses phosphate from internal polyphosphate.

*E. coli* contains two Pit transport systems PitA and PitH [63]. Both systems are 90% identical in amino acid sequence and have similar kinetics. However, they differ in the regulation of their expression; the expression of *pitA* is constitutive under different inorganic phosphate concentrations while that of *pitH* is induced in condition of low phosphate availability [63,73].

### 3.2. The pitH1 and pitH2 Systems in Streptomyces and Corynebacterium glutamicum

Two Pit transport systems have also been identified in the genome of *S. coelicolor* [74] and are present in the genomes of other streptomyces species, e.g., *S. avermitilis* [19,75,76] and *S. tsukubaensis* [14]. These systems were named PitH1 and PitH2 and they contain two transmembrane spanning domains [74]. The PitH1 (332 amino acids) and PitH2 (423 amino acids) of *S. coelicolor* share 44% identity. Upstream of *pitH1* there is a gene named *pap* (encoding a 206 amino acids putative Pit ancillary protein) that is also present in the PitH1 orthologous gene in different *Streptomyces* species and even in the unrelated bacterium *Sin. meliloti* [66,77]. The *pap–pitH1* genes are co-transcribed as a bicistron from a single promoter located upstream of *pap* both in *Sin. meliloti* and *S. coelicolor* (see below, Figure 2). The *pitH2* gene in *S. coelicolor* is located 2.2–2.3 Mb, far from *pitH1*, and is surrounded by genes that are not related to phosphate uptake or metabolism.

PitH1 is the major phosphate transporter in high phosphate concentration conditions (above 3.2 mM) and its expression does not respond to phosphate concentrations during fermentation, whereas *pitH2* expression increases when phosphate is limiting (about 0.01 mM). In a *phoP*-null mutant the *pitH2* promoter is not active and RT-PCR studies showed only a basal level transcription. 

In *S. coelicolor*, PhoP binds to the upstream region of *pitH2*, as shown by electrophoresis mobility assays, but not to the *pap–pitH1* promoter. PhoP protection assays revealed a 64 nt protected region upstream of *pitH2*. This region includes four 11 nt each core Dru’s, and two additional Dru’s with lower nucleotides conservation. Four core Dru’s were protected by very low phoP concentration and, as the PhoP concentration increases, the surrounding ancillary Dru’s were also protected [74]. Four DNA complexes were separated in gel electrophoresis depending on the PhoP protein concentration used in the electrophoretic mobility shift assay (EMSA). The formation of several DNA–PhoP complexes is characteristic of type III promoter regions, with Dru’s having distinct affinity for the PhoP regulator [37,78]. In a *phoP*-null mutant, the *pitH2* promoter is not active, and RT-PCR studies showed only a basal transcription level.

A similar organization has been found in the promoter regions of the *pitH2* in *S. avermitilis*; this strain also contains four DRu’s, two of them that adjust to the consensus PHO boxes and adjacent to them two DRu’s with lower sequence conservation. 

*C. glutamicum* contains a *pitA* and *pitB* that are slightly larger than those of *S. coelicolor.* Noteworthy, the *C. glutamicum pitA* gene is not regulated by the PhoS-PhoR two-component system [74], and is expressed constitutively in the parental strain; *pitB* is a cryptic homologous and does not appear to be transcribed under the experimental conditions used [79,80]. 

### 3.3. PitH Genes in Other Streptomyces Species

In addition to *S. coelicolor* and *S. avermitilis*, we investigated the presence of *pitH* genes in several *Streptomyces* species (*S. clavuligerus*, *S. venezuelae*, *S. tsukubaensis*, and *S. griseus*). In all of them, there are two different *pitH* genes, with exception of *Streptomyces clavuligerus*, which contains three *pitH* genes; the protein encoded by the third gene, *pitH3*, is more similar to PitH2 (55% identical amino acids) than to PitH1 (44%); the *pitH3* gene is located in the megaplasmid pSCL4 (1.8 Mb) suggesting that *pitH* genes might have been mobilized among *Streptomyces* through plasmids transfer. The *pitH1* genes of different *Streptomyces* are 86–90% identical at amino acid levels, and in all *Streptomyces* are preceded by the *pap* gene, encoding a very well conserved 206 amino acids protein. 

The *pitH2* genes are located far from the *pitH1* and the proteins encoded in different *Streptomyces* species are very similar among themselves (72–76% identical amino acids). 

## 4. The *pap-pitH1-pstSCAB-pptA-nudA-ppk* Supercluster 

The location of the phosphate transport genes, and particularly their possible organization in clusters, is of high interest, since it will contribute to understand their expression. We observed that in all *Streptomyces* species, the *pap–pitH1* operon is linked, and in opposite orientation, to the *pstSCAB* cluster and the *ppk* gene encoding the polyphosphate kinase (Figure 2), a fact that has not been reported previously. In addition, the gene SCO4144, downstream of *ppk*, encodes a protein with 67% identity to the phosin PptA that binds metals and polyphosphate [81]. The protein encoded by SCO4144 contains all of the histidine residues characteristics of metal-binding proteins and the arginine residues for sulphate binding found in PptA. The bioinformatic analysis of the SCO4143 protein reveals high similarity to *Streptomyces* nudix hydrolase family proteins involved in dinucleotides hydrolysis and we designated it *nudA*. This finding is interesting since it indicates that *Streptomyces* species have integrated these genes in a short region of the genome. Such organization may improve the assimilation of phosphate, its storage as polyphosphate, and/or the degradation of the latter [82,83,84,85,86]. In contrast to *Streptomyces* sp., in *C. glutamicum* the *pitH1* and *pstSCAB* genes are not linked in a supercluster. 

## 5. Transport and Utilization of Glycerol-3-Phosphate and Glycerophosphodiesters

Phospholipids (e.g., phosphatidylcholine, phosphatidylserine, phosphatidylethanolamine, phosphatidylglycerol) are components of all living beings and therefore they are abundant in some habitats as part of decaying plants and animals. Glycerol phosphodiesters are formed by deacylation of phospholipids and retain a phosphate in carbon-3 of the glycerol. The phosphodiesters are hydrolysed by glycerophosphodiester phosphodiesterases (GDPDs) that cleave the ester bond releasing glycerol-3-phosphate.

Initially it was considered that all glycerophosphodiesters could be hydrolysed extracellularly and the phosphate transported by the well-known phosphate transport systems, but in *E. coli* it was found that mutants defective in the extracellular alkaline phosphatase were still able to utilize glycerol-3-P and glycerol phosphodiesters as sole phosphate sources, indicating that these compounds were transported in their phosphorylated form through the membrane. A gene cluster *ugpBACQ* (for uptake of glycerol-3-P), encoding a typical ABC transporter, was found in *E. coli* [87,88]. Glycerol-3-P is transported without been hydrolysed, whereas phosphodiesters are hydrolysed during transport by the UgpQ membrane associated protein [87]. In addition, *E. coli* contains a separate glycerol-3-phosphate permease, named GlpT, which is part of the *glpQT* operon that encodes also the GlpQ phosphodiesterase [89,90] (Figure 3). 

In the actinobacteria *C. glutamicum*, phylogenetically related to *Streptomyces* species, there is a gene *glpQ* for a phosphodiester phosphodiesterase linked to the *ugp*CBEA operon that encodes an ABC transporter of glycerophosphodiesters [80] (Figure 3). This *ugp* cluster in *C. glutamicum* responds positively to the phosphate limitation in the culture medium.

*Streptomyces* and several other soil dwelling actinobacteria are able to utilize the phosphate of glycerophosphodiesters. However, there are important differences in the utilization of glycerol-3-phosphate and glycerol-phosphodiesters between *E. coli* and the actinobacteria. In *S. coelicolor*, there are seven putative GDPD genes [33,91]. Three of these genes (named *glpQ1* to *glpQ3*) encode secreted GDPDs, whereas the other four genes encode intracellular uncharacterized phosphodiesterases. Gene orthologous to the *glpQ* genes of *S. coelicolor*, encoding proteins with 67 to 80% amino acid identity are disperse in the genomes of several other *Streptomyces* species, including *S. avermitilis*, *S. clavuligerus* and *S. venezuelae*. 

However, we have not found in *S. coelicolor* genome genes for cluster similar to *ugpCBEA* linked to glycerophosphodiester phosphodiesterases genes, in contrast to the organization in *C. glutamicum.* Furthermore, neither *C. glutamicum* nor *S. coelicolor* genomes contain a gene similar to the *E. coli glpT*, which encodes a glycerol-3-P permease (Figure 3). These results suggest that in contrast to *E. coli*, glycerophosphodiesters are not transported as such in *Streptomyces*, but are hydrolysed by extracellular phosphodiester phosphodiesterases that scavenge the phosphate group. 

### Glycerophosphodiester Phosphodiesterases: Molecular Mechanisms of Binding of PhoP to the glpQ Promoters and Regulation by Phosphate and Carbon Sources

In phosphate shift down experiments expression of *glpQ1* and *glpQ2* increased in response to phosphate starvation while expression of *glpQ3* was not altered [17,91]. In a *S. coelicolor phoP* mutant, expression of *glpQ1* and *glpQ2* did not respond to phosphate starvation indicating that the transcriptional response is PhoP-dependent. Both the promoters of *glpQ1* and *glpQ2* show complex structures in relation to phosphate control. There are four direct repeated DRu’s in the promoter of *glpQ1* as revealed by binding of pure PhoP and footprint analysis. In the promoter of *glpQ2* there are five DRu’s, but some of them have low conservation, according to the consensus sequence of Sola-Landa et al. [37], based on their information content [92]. Cooperative binding of two or more PhoP monomers resulted in formation of up to four migration bands in EMSA that correspond to the increasing amounts of the PhoP protein bound to these promoters [91]. The *glpQ1* and *glpQ2* promoters share an organization in which PhoP binds to Dru’s overlapping the -35 promoter region as concluded from identification of the transcription start site. Noteworthy, both inorganic phosphate and glycerol-3-phosphate regulate expression of *glpQ1* and *glpQ2* promoters in *S. coelicolor* [91]. An open question is whether the glycerol-3-phosphate molecule is internalized into the cells as occurs in *E. coli* [87,88]; the absence in *Streptomyces* species of a *ugp* cluster encoding an ABC transporter for glycerol-3-phosphate suggests that this compound is hydrolysed before been transported.

Expression of the *glpQ1* and *glpQ2* is also regulated by carbon sources; this is not surprising since cleavage of the phosphodiester bond by the phosphodiesterases releases serine, ethanolamine, inositol, and other amines or alcohols. Interestingly, Santos-Beneit et al. [91] observed that expression of *glpQ1* was repressed by both serine and inositol whereas expression of *glpQ2* was not affected by these carbon sources, although it was induced by the preferred carbon sources glucose, fructose, and glycerol. The induction of *glpQ2* by these compounds is independent of the phosphate regulation that still is fully active when the cells are grown in these carbon sources, suggesting an accumulative regulation by different effectors [2].

## 6. Utilization of Sugar Phosphates and Nucleotides: Dephosphorylation or Transport of Phosphorylated Compounds?

So far, we reviewed the utilization of glycerol-**3**-P in *E. coli* and other bacteria. In addition, in different habitats there are numerous sugar phosphates and nucleotides that provide phosphate and carbon sources for growth; sometimes these compounds are released by lysed cells of confluent cultures and can be utilized as nutrient by the surviving cells [93,94]. An important question is if intact nucleotides or sugar phosphates are transported in their phosphorylated form or whether they are dephosphorylated before being taken up.

In *E. coli* and *Salmonella*, some sugar phosphates are transported by the *uhpABCT* system for uptake of hexose phosphates [95] that includes the permease *uhpT* [96,97]. As indicated above, *C. glutamicum* contains a glycerol-3-phosphate transporter [80], but does not contain a glucose-6-phosphate transporter; importantly, bioinformatic analysis reveals that non-clustered *uhpT* genes encoding homologous transporters, occur in *Corynebacterium lactis* and in several pathogenic corynebacteria including *Corynebacterium diphtheriae* and *Corynebacterium pseudotuberculosis* (about 49% amino acids identity with the *E. coli* UhpT protein), but are not present in non-pathogenic *Corynebacterium sp.* or in *Arthrobacter sp.* It is likely that pathogenic corynebacteria have adapted to transport hexose phosphates obtained from animal tissues of the infected host.

In *Streptomyces*, glucose 6-phosphate is used both as a carbon and phosphate source. Tenconi et al. [98] tested the effect of sugar phosphates, including glucose-6-P, glucosamine-6-P, fructose-6-P and fructose 1,6 bisphosphate on differentiation of *S. coelicolor* and *S. lividans*, and their effect on the production of antibiotics by these species. All of these sugar phosphates, but not their dephosphorylated forms, delayed the differentiation and the production of secondary metabolites. Noteworthy, the same effect exerted by the sugar phosphate was observed when the corresponding amounts of inorganic phosphate was added. The utilization of these sugar phosphates was PhoP-dependent and was not observed in PhoP defective mutants [98]. These authors did not find evidence of transport of phosphorylated glucose and proposed that glucose-6-P is dephosphorylated by extracellular or cell membrane anchored alkaline phosphatases [12] prior to its transport into the cells. Genes orthologous to the *E. coli* o *C. diphtheriae uhp* have not been found by bioinformatic analysis in *Streptomyces* sp. genomes. Independently of whether sugar phosphate transport exist, it is possible that these sugar phosphates trigger a signalling cascade in *Streptomyces* sp. by interacting with membrane receptors. 

Regarding the utilization of nucleotides in early studies, Martín and Demain [99] analysed the effect of a variety of nucleotides, nucleosides, purine, and pyrimidine bases on the biosynthesis of candicidin in *S. griseus* by measuring the incorporation in candicidin of labelled precursors. All of the nucleotides have an inhibitory effect on candicidin biosynthesis, including cAMP, but the nucleosides or the free bases had no effect. These results suggest that the effect of the nucleotides on candicidin biosynthesis in *S. griseus* is probably due to the phosphate released by hydrolysis of the nucleotides. Indeed, the pure alkaline phosphatase of that strain was found to have a broad substrate specificity including the hydrolysis of nucleotides [11]. In addition, a PhoP-regulated extracellular 5′nucleotidase (SCO4152) has been described in *S. coelicolor* [18].

In *S. coelicolor* addition of extracellular ATP has a dose-dependent effect up to 10 mM ATP, increasing the production of actinorhodin by stimulating expression of *actII-orf4* regulatory gene, but ATP was repressive at concentrations above 10 mM [100]. These authors did not study the transport of ATP and suggested that ATP might have a regulatory role by interacting with a specific receptor in the cell membrane. The differential effect of ATP on antibiotics formation in *S. coelicolor* may be also related to the high intracellular ATP concentration in this *Streptomyces* strain [101]. 

## 7. Transport and Utilization of Phosphonates

Another phosphate source that might be utilized by microorganisms are phosphonates. Phosphonates are organic compounds that contain a phosphorous linked covalently to a carbon atom (C–P) in the skeleton of the molecule. The phosphonates characteristic C–P bond is energetically very stable and difficult to hydrolyse. There are hundreds of phosphonates among the metabolites of all living beings, such as the phosphonolipids and other natural compounds produced by microorganisms (e.g., antibiotics as fosfomycin or herbicides as bialaphos). In addition, since the arrival of the industrial era, thousands of man-made compounds containing the C–P bond have been synthesized. Many of these compounds are xenobiotics and present serious degradation problems. Therefore, the metabolism of phosphonates has great interest in ecology and bioremediation [102]. However, only bacteria and lower eukaryotes have the ability to hydrolyse the C–P bond in these compounds to utilize phosphate as nutrients [62,103]. Simple phosphonates, such as methylphosphonic acid or ethylphosphonic acid, have been used routinely to test the degradative ability of microorganisms. Glyphosate is also a convenient substrate because it is degraded to sarcosine (N-methylglycine) that is easily detected in chemical analysis. Early tests on phosphonate degradation were performed with *E. coli*, *Klebsiella aerogenes* and *Pseudomonas sp*. [104,105], but also several later studies include the ability to degrade phosphonates by gram positive bacteria and fungi [106,107,108].

Prior to investigating phosphonate transport and utilization genes in *Streptomyces*, it is necessary to understand the arrangement of these genes and their biochemical function in *E.coli.* The phosphonate gene cluster in *E. coli* is formed by a set of 14 genes for phosphonate transport and utilization, namely *phnCDEFGHIJKLMNOP* [109] (Figure 4). Mutants disrupted in *phn* genes were unable to degrade simple phosphonates [62,104]. The fourteen genes cluster is expressed as a single operon from a promoter located upstream of *phnC*. 

Additional analysis of the cluster concluded that the first three genes *phnCDE* constitute an ABC system for phosphonate transport closely similar to the *phnDCE* cluster of *M. smegmatis* that in this bacterium behaves as a high affinity phosphate transporter [56] (see above) (Figure 4). The hydrophilic PhnD protein, is a phosphonate-binding protein, and is located in the periplasmic space as occurs with the homologous protein of *Sin. meliloti* [61]. The PhnC protein has an ATP binding sequence and is similar to nucleotide-dependent permeases such as the PstB protein (a member of the PstSCAB system) for phosphate transport. The hydrophobic PhnE protein is a membrane integral permease with homology to the phosphate permease PstA. The set of genes *phnG* to *phnM* encode essential components of the C–P lyase complex and enzymes for phosphonate degradation; while PhnN, that has phosphokinase activity, and PhnP appear to be accessory proteins. Finally, PhnO is an amino alkyl phosphonate N-acetyltransferase not essential for phosphonate cleavage and PhnF is a regulatory protein. Gene clusters similar to those of *E. coli* have been found in several enterobacteriaceae and rhizobiaceae, particularly in *Sin. meliloti* but there is no report of these clusters in Gram positive bacteria, with exception of the high affinity *phnDCE* cluster in *M. smegmatis* (see above). Importantly, expression of the *phnCDEFGHIJKLMNOP* cluster in *E. coli* is regulated by the inorganic phosphate concentration in the medium and the promoter upstream of *phnC* gene contains an 18 nucleotide PHO box similar to that recognized by the PhoB regulators in other Gram-negative bacteria.

### 7.1. Phosphonate Transport and Utilization in Arthrobacter and Corynebacterium Species

Early studies on utilization of phosphonates by actinobacteria were made in *Arthrobacter* strains. *Arthrobacter* species are bacteria that proliferate in a variety of habitats and have a wide ability to degrade different carbon substrates, including phosphonates [110,111]. One strain characterized as *Arthrobacter* sp. GLP-1 was selected since it was able to degrade glyphosate and utilize alkyl phosphonic acids as the only phosphate source [106,110]. Utilization of simple or complex phosphonates in both Gram negative and *Arthrobacter* sp. require a growth adaptation time indicating that the cells do not have a constitutive transport and enzymatic system for the degradation but respond to the presence of the phosphonates as inducers. This degradation system has not been studied at the biochemical or molecular genetic level.

Using *E. coli* phosphonate transport genes *phnCDE* as probe many homologous genes are detected in *Arthrobacter* species genomes, either forming clusters or in separated locations. With respect to the genes for phosphonate degradation, three genes for subunits of the C–P lyase, *phnL* and *phnK* and the gene *phnM* for the α-D-ribose 1-methylphosphonate 5-triphosphate diphosphatase, enzyme essential in phosphonate degradation, are present in many *Arthrobacter* strains; however, other genes for other subunits of the C–P lyase complex, namely *phnIHG*, or the gene *phnN* encoding a phosphokinase required for phosphonate degradation are not present in most *Arthrobacter* species. Only the sequenced genomes of three strains, *Arthrobacter glacialis*, *Arthrobacter alpinus* and *Arthrobacter subterraneus* (Figure 4), contain all these genes (40–70% amino acids identity to those of *E. coli*). For example, *A. glacialis* contains a cluster *phnDCE* and a separated cluster *phnFGHIJKLMN* and similar results were observed in the other two strains, but the type strain, *Arthrobacter globiformis*, does not contain genes for any of the C–P lyase subunits.

*Corynebacterium* species are phylogenetically close to *Arthrobacter* sp. and *Streptomyces* sp. In saprophytic corynebacteria there are many species that have homologous to the transport genes *phnC*, *phnD* and *phnE*, including the model organism *Corynebacterium glutamicum*, an industrial glutamic acid producer, which has a *phnEC* cluster and a separate *phnD* gene. Some genes for phosphonate degradation, as those for the subunits PhnL and PhnK, are present in many *Corynebacterium* strains, but genes for the other C–P lyase subunits (*phnG*, *phnH*, *phnI*, *phnJ*), or for enzymes required for phosphonate degradation (*phnM*, *phnN*), are rare, and only *Corynebacterium doosanense* appears to have a whole set of genes for phosphonate transport and degradation, although separately located (Figure 4).

### 7.2. The Ability to Utilize Phosphonates Is Restricted to Some Streptomyces sp. 

Numerous organic compounds and phosphate sources have been tested for decades in the search of optimal media to produce new bioactive compounds by *Streptomyces* strains. However, detailed studies on phosphonate utilization are limited to the *Streptomyces* sp. described by Obojska et al. [112]. Using strains isolated from a sewage treatment plant in Wroclaw (Poland) these authors isolated a number of colonies that were able to grow on different phosphonates. Two strains identified as *Streptomyces*, based on their morphological characteristics, the presence of diaminopimelic acid in the cell wall and the type of phospholipids in the membrane, were named *Streptomyces* StA and *Streptomyces* StC. These strains were studied by their ability to grow on three phosphonates, namely 2-aminoethanephosphonic acid, glyphosate, and phosphonoacetic acid as sole phosphate or carbon source. *Streptomyces* StC, a strain resistant to 100 mM glyphosate, was particularly able to degrade glyphosate releasing to the culture medium inorganic phosphate, sarcosine and glycine as degradation products. This strain utilizes glyphosate even in the presence of moderate concentrations of phosphate suggesting that the C–P lyase system of this strain is not strictly repressed by the inorganic phosphate in the medium. Cell-extracts of the second isolate, *Streptomyces* StA, degrade in vitro 2-aminoethanephosphonic acid suggesting the involvement of a C–P lyase in the extract. A minimal growth time of 10 h on 2-aminoethanephosphonic acid was required to observe this degrading activity in cell extracts. No further biochemical or molecular genetic analysis of the phosphonate utilization by these two strains are available so far.

Genes homologous to the *phn* genes of *E. coli* are rare in *Streptomyces* species. One strain, *Streptomyces cavourensis*, has three clusters: *phnDCEE*, *phnCE*, and an incomplete cluster *phnFGHIKLM*. Using these *phn* genes as probe to search in other *Streptomyces* genomes we observed that although genes homologous to the transporter genes *phnC* and *phnE* occur usually close together in many species, the gene *phnD* encoding the phosphonate-binding protein is less frequent. However, the PhnD protein is essential for phosphonate transport. Genes for phosphonate degradation (C–P lyase components) are very rare in *Streptomyces* sp. Bioinformatic analysis reveals that only a few *Streptomyces* species have the genetic information to degrade phosphonates, namely *Streptomyces purpurogeneiscleroticus* that has a *phnDCE* cluster and a separate *phnMLKIHFEEDCNKG* cluster (Figure 4), where F is a GntR-type regulator, or *Streptomyces rhizosphaericus* that has two putative clusters, *phnMLJIHFCED* and *phnCEG.* Other *Streptomyces* species contain incomplete clusters for phosphonate degradation, most times lacking some of the essential genes for the C–P lyase subunits *(phnGHIJKL)* and all of them lack the genes *phnO* and *phnP*, which in *E. coli* encode proteins non-essential for phosphonate degradation. In summary, although *phn* genes are present in some *Streptomyces* sp. the characterization of genes for the subunits of the C–P lyase is required in order to understand whether the phosphonates are really hydrolysed. This lack of information is really surprising since several herbicides extensively used in agriculture are phosphonates, including bialaphos and glyphosate. Noteworthy, many of the strains containing genes for phosphonate degradation are plant endophytes or isolates from soil roots or from mangroves, suggesting that phosphonate utilization gene clusters have evolved to utilize phosphonates in some plant rhizosphere. Some *Streptomyces* sp. are able to synthesize phosphonates as natural products. This is the case of *Streptomyces hygroscopicus* and *Streptomyces viridochromogenes* that produces bialaphos and *Streptomyces fradiae* that produces fosfomycin [113,114]. These strains secrete these phosphonate compounds, but they are unable to degrade them since they are not transported back; this is not surprising since many *Streptomyces* secrete toxic secondary metabolites, but they do not transport them back to the cell to avoid suicide [115]. Secretion of fosfomycin and other antibiotics takes place mainly through the major facilitator superfamily transport systems [116], which are different from the phosphonate transport systems.

In summary, in contrast of the linkage of all phosphonate utilization genes in *E. coli*, the organization in *Streptomyces* is different. Frequently the *phnCDE* genes occur, but they are not linked to the *phnFGHIJKLMN* genes. This finding points to the possibility that, as occurs in *M. smegmatis* the *phnCDE* genes that encode the phosphonate/phosphate transporter ABC transporter might have evolved separately and they serve as inorganic phosphate (rather than phosphonate) transporters.

## 8. Conclusions

Taking into account all available evidence, it is concluded that actinobacteria in general lack some of the phosphate transport systems that exist in *E. coli* and other Enterobacteria. Only the high affinity PstSCAB and the low affinity duplicate Pit (PitH1–PitH2) systems are universally conserved in all studied actinobacteria including *Arthrobacter*, *Mycobacterium*, *Corynebacterium*, and *Streptomyces* species. An important observation is that three genes clusters involved in phosphate metabolism are associated in a short region of *Streptomyces* genomes, forming a phosphate metabolism supercluster. There are important differences in the ability of bacteria to transport and grow on distinct phosphate substrates; this is a normal behaviour since phosphate in nature is present in quite different forms in distinct habitats and the bacteria face the challenge to utilize the phosphate form that is available. In contrast to pathogenic bacteria, most *Streptomyces* species are soil habitants and, therefore, have powerful systems to scavenge phosphate from soil. Genes for *uhp* hexose transport exist in some pathogenic corynebacteria, as it is the case of *Corynebacterium lactis*, but is absent in the *C. glutamicum* type strain. It is likely that this pathogenic corynebacteria have developed a hexose phosphate transporter required for utilization of sugar phosphate from the host tissues in which they live. *Streptomyces* species do not contain an *uhp* gene cluster and there is no evidence (despite several research articles) that sugar phosphates are transported in their phosphorylated form into the cells. There is evidence that sugar phosphates and nucleotides are hydrolysed by extracellular cell-membrane associated alkaline phosphatases and nucleotidases and the phosphate is then transported into the cell. A few *Arthrobacter* and *Streptomyces* are able to utilize phosphonates, which contain the very stable C–P bond, but this appears to be restricted to *Streptomyces* saprophytic species that live associated with plant roots and mangroves. Other *Streptomyces* species lack phosphonate degradation genes, although they have the *phnCDE* transporter system that may be used as an alternative phosphate, rather than phosphonate, transporter. *Streptomyces* species have a very diverse degradative and biosynthetic metabolism and we cannot exclude that phosphonates may be degraded by enzyme systems different from the well-known system in Enterobacteria. Additional biochemical research is required to clarify this obscures points. In summary, *Streptomyces* species are less versatile than Enterobacteria, although some plant-rhizosphere *Streptomyces* species have the genetic capability to degrade phosphonates.

## Figures and Tables

**Figure 3 ijms-22-01129-f003:**
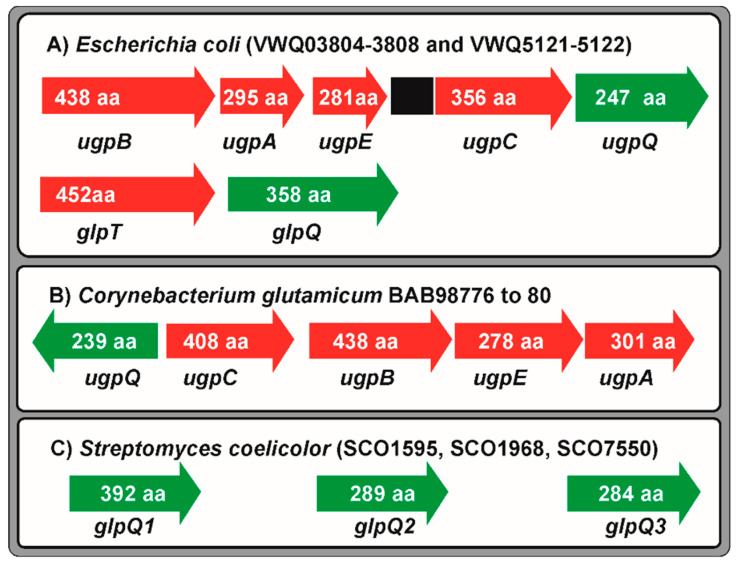
Gene clusters for glycerol-3-phosphate and phosphodiesters uptake in *E. coli* and Actinobacteria. (**A**) Cluster *ugpBAECQ* and *glpQT* in *E. coli.* (**B**) Cluster *upgQABE* in *C. glutamicum*. (**C**) The non-clustered extracellular glycerophosphodiester phosphodiesterases of *S. coelicolor*. Note that *Streptomyces* species lack the *ugp* cluster. The accession number of the genes and the number of amino acids is indicated. Transporter genes are shown in red. Glycerophosphodiester phosphodiesterases genes are shown in green. Genes not related with glycerol-**3**-P or phosphodiesters uptake are indicated in black.

**Figure 4 ijms-22-01129-f004:**
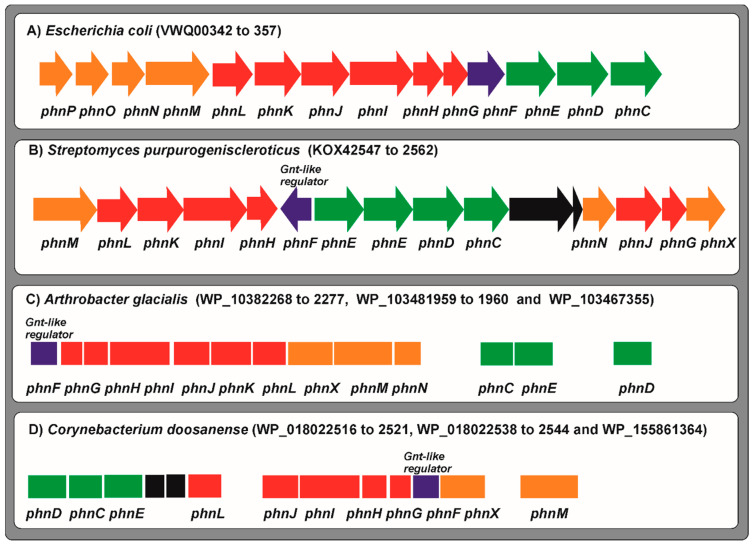
Clusters of genes for phosphonates transport and degradation in *E. coli* and Actinobacteria. The genes for uptake of phosphonates are shown in green; genes for the structural subunits of the C–P lyase are indicated in red. Genes involved in further metabolism of phosphonates are shown in orange, the *phnX* gene is annotated as a not specified phosphonates metabolism gene. The regulatory genes are highlighted in dark blue. Genes not related to phosphonates uptake or metabolism are shown in black. The cluster of genes of *E. coli* (**A**), *Streptomyces purpurogeniscleroticus* (**B**), *Arthrobacter glacialis* (**C**) and *Corynebacterium doosanense* (**D**) are indicated with their accession number in the Databases. The name of the genes is indicated below.

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
