# Peer review of "Molecular Mechanisms of Phosphate Sensing, Transport and Signalling in Streptomyces and Related Actinobacteria"

_ijms, 2021, doi:10.3390/ijms22031129_

Round 1
Reviewer 1 Report
In the paper entitled « Molecular mechanisms of phosphate sensing, transport and signaling in Streptomyces and related actinobacteria” the authors, Juan F. Martín and Paloma Liras, review what is currently known on uptake of inorganic phosphate, organophosphate and phosphonates in Streptomyces and related actinobacteria often in comparison with model Gram – strain, E. coli. The paper is rather well structured and written and easy to follow. However some references related to the topic are missing.
Minor comments and questions
P1: Abstract: It might be preferable to say “seems less versatile” than “is less versatile”.
P2: Introduction: Suggestion of a slightly different formulation “Phosphate is a crucial element of cell metabolism, since it is constitutive of nucleic acids, phospholipids and teichoic acids. It also plays important roles in oxidative phosphorylation/respiration that generates highly phosphorylated nucleotides and in signaling cascades that involves phosphorylated proteins [1,2].
P2 …” by the intracellular phosphate concentration that reflects….”
P2 Besides ref15, it would be nice to quote also https://journals.plos.org/plosone/article?id=10.1371/journal.pone.0126221
P2 It is somehow paradoxical that high and low affinity transport systems are induced in condition of phosphate limitation. Their threshold of induction is probably different.
P3 The formulation found throughout the manuscript “The system in Streptromyces and most other bacteria is organized as an ATP-binding cassette …” is rather unusual and surprising since one usually speaks of ABC transport system.
P3” In addition to regulation by the PhoP master regulator, formation of the PstS protein was highly increased in a mutant deficient in polyphosphate kinase [33] which catalyzes in vivo, in condition of phosphate limitation, the regeneration of ATP from ADP and polyphosphates
Please quote: https://onlinelibrary.wiley.com/doi/full/10.1046/j.1365-2958.2002.02557.x and https://www.ncbi.nlm.nih.gov/pmc/articles/PMC5427975/
and add.
This is most likely due to the up-regulation of phoR/phoP observed in the ppk mutant strain. https://pubmed.ncbi.nlm.nih.gov/16385057/
P3 Is the expression of PstS repressed in the presence of glucose?
P3 Suggestion of modification “The pstSCAB operon of S. lividans is expressed as a single transcription unit but levels of the pstS gene transcript were higher than those of other genes of the operon [31]. This indicated that PstS is co-transcribed with the other genes of the operon but is also transcribed alone from its own promoter (20). Interestingly the presence of transcription termination sites between the pstS and pstC genes was noted.”
The authors wrote “In S. lividans deletion of the region encoding the PHO boxes decreased the expression of the pstSCAB operon but allows a remaining basal level of expression (about 10%) that was controlled by the carbon source [31].”
Do the authors speak about the expression of the whole operon or of PstS only?
P 3 Concerning the mechanism of induction of the PhoR-PhoP signaling cascade. The authors mentioned two hypothesis but a third hypothesis also exist
The gene phoU is located in divergence of the phoR/phoP operon. PhoU-like proteins were shown to be metalloproteins containing two multinuclear iron clusters (https://pubmed.ncbi.nlm.nih.gov/15716271/° In Streptomyces, the gene located downstream of phoU, mtpA, encodes a 79-aa-long cysteine-rich protein belonging to the family of metallothionein proteins. These two metal binding proteins might be involved in the sensing of and/or resistance to oxidative stress (https://pubmed.ncbi.nlm.nih.gov/12022471/).
Furthermore PhoU is known to have a positive impact of the level of PhoR/PhoP expression (https://jb.asm.org/content/jb/188/2/677.full.pdf). It was thus proposed that since phosphate limitation is known to correlate with enhanced oxidative stress (https://pubmed.ncbi.nlm.nih.gov/11251823/), oxidative stress might lead to some PhoU conformational changes conferring to PhoU the ability to stimulate kinase PhoR auto phosphorylation.
Oxidative stress linked to phosphate limitation that results into the activation of the oxidative metabolism as demonstrated in https://www.ncbi.nlm.nih.gov/pmc/articles/PMC5427975/ is proposed to be the real signal triggering the expression of PhoR/PhoP.
P4 Concerning the “Teichoic acid hypothesis”. The fact that it exists in Streptomyces a cluster for teichoic acid biosynthesis regulated by phosphate does not constitute at all a support for the hypothesis ! Please modify this sentence.
At last it would be nice to mention somewhere in this part that it was recently reported that the expression of genes of the Pho regulon involved in Pi scavenging and uptake is strongly reduced in Streptomyces coelicolor compared to S. lividans
https://www.nature.com/articles/s41598-020-65087-w
P 5 The terms discriminate/discrimination do not seem correct.
Rather than saying “A comparison between phosphate and arsenate discrimination by PstS proteins indicates that they are able to discriminate about 500 to 700-fold in favor of phosphate [43]”, it is probably better to say “The affinity of the PstS protein for phosphate is 500 to 700 higher than that for arsenate”
“ A PstS protein with high affinity for arsenate has been crystalized….”
“ The arsenate anion is 4% larger than the phosphate anion. Its accommodation in PstS requires the distortion of this low barrier hydrogen bond.”
P5 Is the nature of the sugars involved in PstS glycosylation known?
Does PstS glycosylation regulates PstS activity and anchorage to the membrane?
P6 “In summary, expression of the pstS gene in S. coelicolor drastically increases in response to phosphate starvation and, therefore, the protein is accumulated in the cell membrane and under adequate conditions is released.”
Does the release of PstS impairs Pi uptake?
P6 “The PhnF protein represses expression of the phnDCE cluster by binding to two sides in the bidirectional promoter regions [52].”
What do the authors mean by “two sides”?”
P7 the authors probably mean “… and uses phosphate from internal polyphosphate degradation” and not phosphite.
P7 “…: the expression of pitA is constitutive under different inorganic phosphate concentrations while that of pitH is induced in condition of low phosphate availability [58, 68].”
P8 Figure 2 legend
The gene sco4144 located behind ppk encodes a phosin bearing a CHAD domain that is considered as a polyphosphate binding module. https://pubmed.ncbi.nlm.nih.gov/31183865/
This enzyme is considered as an accessory factor involved in the upkeep of polyP in a conformation suitable for its use by Ppk and/or its degradation by exopolyphosphate phosphatases.
Sco4143 encodes a protein of the nudix hydrolase family bearing similarities with enzymes involved in the cleavage of diadenosine hexaphosphate (Ap6A) into two molecules of ATP.
Furthermore it would be nice to provide SCO names for the genes mentioned.
What does pap (SCO4137) located upstream of pitH1 stand for?
Annotation suggest that it might be related to PhoU ?
P9 “This finding is interesting since it indicates that Streptomyces species have integrated these genes in a short region of the genome [76]. Such organization may benefit to the assimilation of phosphate, its storage as polyphosphate and/or the degradation of the latter”
In fact nobody knows what drives polyphosphate biosynthesis is Streptomyces as in most other micro-organisms but PhoU seems to be involved
https://pubmed.ncbi.nlm.nih.gov/12147514/
https://jb.asm.org/content/199/18/e00143-17
P10 “These results suggest that in contrast to E. coli, glycerophosphodiesters are not transported as such in Streptomyces but are hydrolyzed by extracellular phosphodiester phosphodiesterases that scavenge the phosphate group.”
P11 “Independently of whether sugar phosphate transport exist, it is possible that these sugar phosphates trigger a signaling cascade in Streptomyces sp. by interacting with membrane receptors.” That is pure speculation…
P11 “In S. coelicolor addition of extracellular ATP have a dose-dependent effect up to 10 mM ATP, increasing production of actinorhodin by stimulating expression of actII-orf4 regulatory gene but it was repressive at higher concentration [92].”
This observation can perhaps be related to the fact that S. coelicolor has a 2 to 3 fold higher intracellular ATP concentration than S. lividans (https://www.ncbi.nlm.nih.gov/pmc/articles/PMC5427975/).
High ATP concentration can signals active oxidative metabolism as well as perhaps also growth slow down (ATP generation exceeding ATP needs for anabolism)
P12 “Mutants disrupted in the phn gene cluster were unable to degrade simple phosphonates [57, 95].” This sentence is unclear please re-formulate.
P14 That is unclear whether the presence / absence of phn genes have been correlated or not with the ability to degrade phosphonates ?
P15 The sentence “An important observation is that three genes cluster involved in phosphate metabolism are associated in a short region of Streptomyces genomes forming a phosphate metabolism supercluster.” is repeated in line 649-641 and in lines 657-659.
P15 line 664 Do plants produce numerous C-P compounds?
Author Response
In the paper entitled « Molecular mechanisms of phosphate sensing, transport and signaling in Streptomyces and related actinobacteria” the authors, Juan F. Martín and Paloma Liras, review what is currently known on uptake of inorganic phosphate, organophosphate and phosphonates in Streptomyces and related actinobacteria often in comparison with model Gram – strain, E. coli. The paper is rather well structured and written and easy to follow. However, some references related to the topic are missing.
We acknowledge the comments of the reviewer. Some of the suggested references correspond to publications on Streptomyces metabolism and stress rather than to articles on phosphate transport. According to the instructions of the editor for the special issue on phosphate transport in bacteria the scope of the article should be focused on phosphate transport. As indicated in the Introduction we have not included in this review information on Streptomyces metabolism. In any case we have accepted all the comments and references suggested by the reviewer and have modified the manuscript accordingly.
Minor comments and questions
Page 1: Abstract: It might be preferable to say “seems less versatile” than “is less versatile”.
Answer: Corrected as suggested by the reviewer in line 39, it says: ”In summary, Streptomyces and related actinobacteria seem to be less versatile in phosphate transport systems than Enterobacteria”
Page 2
a) Introduction: Suggestion of a slightly different formulation “Phosphate is a crucial element of cell metabolism, since it is constitutive of nucleic acids, phospholipids and teichoic acids. It also plays important roles in oxidative phosphorylation/respiration that generates highly phosphorylated nucleotides and in signaling cascades that involves phosphorylated proteins [1,2]. P2 …” by the intracellular phosphate concentration that reflects….”
Answer: The sentence has been slightly changed as suggested, but we do not see any essential difference between the wording of the original manuscript and the suggestion of the reviewer. In lane 45 is stated: “Inorganic phosphate (Pi) is a key nutrient in cell metabolism, since it is constitutive of nucleic acids, phospholipids, theicoic acids, membranes, highly phosphorylated nucleotides and phosphorylated proteins. It participates in the respiratory chain and plays an important role in the signalling cascades [1,2]”.
b) Besides ref.15, it would be nice to quote also (Smirnov et al 2015).
Answer: In lane 72, in addition to the ref 15, the reference of Smirnov et al., 2015 has been added
c) It is somehow paradoxical that high and low affinity transport systems are induced in condition of phosphate limitation. Their threshold of induction is probably different.
Answer: The sentence in line 92 has been changed to “…high affinity and low affinity phosphate transport systems which are all of them induced by phosphate limitation, although they may respond to different phosphate threshold for induction”.
Page 3
The formulation found throughout the manuscript “The system in Streptromyces and most other bacteria is organized as an ATP-binding cassette …” is rather unusual and surprising since one usually speaks of ABC transport system.
Answer: The designation ABC-binding cassete has been changed to ABC transporter in the whole text
In addition to regulation by the PhoP master regulator, formation of the PstS protein was highly increased in a mutant deficient in polyphosphate kinase [33] which catalyzes in vivo, in condition of phosphate limitation, the regeneration of ATP from ADP and polyphosphates. Please quote: https://onlinelibrary.wiley.com/doi/full/10.1046/j.1365-2958.2002.02557 and https://www.ncbi.nlm.nih.gov/pmc/articles/PMC5427975/ and add “This is most likely due to the up-regulation of phoR/phoP observed in the ppk mutant strain. https://pubmed.ncbi.nlm. nih.gov/16385057/
Answer: In line 121, the references to Esnault et al 2017 and Chouayekt and Virolle 2002 have been added, as well as the sentence in line 122 “This is most likely due to the up-regulation of phoR/phoP observed in the ppk mutant (Ghorbel et al 2006).”
In addition to regulation by the PhoP master regulator, formation of the PstS protein was highly increased in a mutant deficient in polyphosphate kinase [33] which catalyzes in vivo, in condition of phosphate limitation, the regeneration of ATP from ADP and polyphosphates
Answer: Modified as suggested (line 121)
Is the expression of PstS repressed in the presence of glucose?
Answer: There are no detailed studies on the effect of glucose on the transcription of the PstSCAB operon.
Suggestion of modification “The pstSCAB operon of lividansis expressed as a single transcription unit but levels of the pstS gene transcript were higher than those of other genes of the operon [31]. This indicated that PstS is co-transcribed with the other genes of the operon but is also transcribed alone from its own promoter (20). Interestingly the presence of transcription termination sites between the pstS and pstC genes was noted.” Do the authors speak about the expression of the whole operon or of PstS only?
Answer: In line 132, modified as follows “In S. lividans deletion of the region encoding the PHO boxes decreased the expression of the pstSCAB operon but allows a remaining basal level of expression (about 10%) that was controlled by the carbon source [31].”
We refer to the expression of the entire pstSCAB operon as indicated by Esteban et al.
Concerning the mechanism of induction of the PhoR-PhoP signaling cascade. The authors mentioned two hypothesis but a third hypothesis also exist. The gene phoU is located in divergence of the phoR/phoP operon. PhoU-like proteins were shown to be metalloproteins containing two multinuclear iron clusters (Liu et al 2005) In Streptomyces, the gene located downstream of phoU, mtpA, encodes a 79-aa-long cysteine-rich protein belonging to the family of metallothionein proteins. These two metal binding proteins might be involved in the sensing of and/or resistance to oxidative stress (Coyle et al Metallothionein)
Furthermore PhoU is known to have a positive impact of the level of PhoR/PhoP (Ghorbel et al 2006). It was thus proposed that since phosphate limitation is known to correlate with enhanced oxidative stress (Moreau et al 2001) oxidative stress might lead to some PhoU conformational changes conferring to PhoU the ability to stimulate kinase PhoR auto phosphorylation.
Oxidative stress linked to phosphate limitation that results into the activation of the oxidative metabolism as demonstrated in Esnault et al 2017/ is proposed to be the real signal triggering the expression of PhoR/PhoP.
A third hypothesis make also explain the observed finding, The gene phoU is located in divergence of the phoR/phoP operon. PhoU-like proteins were shown to be metalloproteins containing two multinuclear iron clusters (Liu et al 2005) In Streptomyces, the gene located downstream of phoU, mtpA, encodes a 79-aa-long cysteine-rich protein belonging to the family of metallothionein proteins. These two metal binding proteins might be involved in the sensing of and/or resistance to oxidative stress
Answer: A new sentence has been added (lines 186-192) as follows: “A third hypothesis also possible: the phoU and the adjacent mtpA located downstream of phoU encode metalloprotein might be involved in the sensing or resistance to oxidative stress (Coyle et al 2002; Moreau et al. 2001). Since PhoU is known to have a positive effect on expression of the phoR-phoP operon (Ghorbel 2006, Martin-Martin et al 2017) it is possible that the oxidative stress might lead to a modified PhoU conformational change conferring to PhoU the ability to stimulate kinase PhoR auto phosphorylation.Studies on conformational changes of PhoU are required to support this hypothesis”
Page 4
- Concerning the “Teichoic acid hypothesis”. The fact that it exists in Streptomyces a cluster for teichoic acid biosynthesis regulated by phosphate does not constitute at all a support for the hypothesis! Please modify this sentence. At least it would be nice to mention somewhere in this part that it was recently reported that the expression of genes of the Pho regulon involved in Pi scavenging and uptake is strongly reduced in Streptomyces coelicolor compared to lividans
Answer: The text has been changed to “this correlates well with the …” (line 180). The reference to Millan Oropeza has been included later (line 527)
Page 5
- The terms discriminate/discrimination do not seem correct. Rather than saying “A comparison between phosphate and arsenate discrimination by PstS proteins indicates that they are able to discriminate about 500 to 700-fold in favorof phosphate [43]”, it is probably better to say “The affinity of the PstS protein for phosphate is 500 to 700 higher than that for arsenate”
Answer: The term discrimination is used in the original article (Elias et al., 2012) in which the discrimination of As versus Phosphate was studied. In any case we have changed the text in line 212 as suggested.
- A PstS protein with high affinity for arsenate has been crystalized….”
Answer: In line 221, the sentence has been changed to “A PstS protein with higher affinity for arsenate has been crystalized from Clostridium perfringens “
- The arsenate anion is 4% larger than the phosphate anion. Its accommodation in PstS requires the distortion of this low barrier hydrogen bond.”
Answer: The sentence has been changed in lines 223 to “The arsenate anion is 4% larger than the phosphate anion; its accommodation in PstS requires the distortion of this low barrier hydrogen bond allowing its discrimination by distorting this low barrier hydrogen bond”.
- Is the nature of the sugars involved in PstS glycosylation known?
Does PstS glycosylation regulates PstS activity and anchorage to the membrane?
Answer: Yes, the sugars attached during glycosylation are mannose and mannose oligosaccharides (Wehmeier et al 2009). It is unknown if the glycosylation of PstS affects the anchorage to the membrane, but probably glycosylation is important for activity since the released protein does not bind phosphate
Page 6
- In summary, expression of the pstSgene in coelicolor drastically increases in response to phosphate starvation and, therefore, the protein is accumulated in the cell membrane and under adequate conditions is released.” Does the release of PstS impairs Pi uptake?
Answer: Yes, the released PstS does not bind phosphate as indicated in the original text (now lanes 14 and 255), and therefore decreases Pi transport.
- “The PhnF protein represses expression of the phnDCE cluster by binding to two sides in the bidirectional promoter regions [52].” What do the authors mean by “two sides”?”
Answer: Sites are nucleotide sequences. We have changed the sentence in line 286 to “The PhnF protein represses expression of the phnDCE cluster by binding to two nucleotide sequences in the bidirectional promoter regions”
Page 7
- the authors probably mean “… and uses phosphate from internal polyphosphate degradation” and not phosphite.
Answer: It was a typographical error. Changed to phosphate in line 314
- “…: the expression ofpitA is constitutive under different inorganic phosphate concentrations while that of pitH is induced in condition of low phosphate availability [58, 68].”
Answer: Changed in line 317 to “However, they differ in the regulation of their expression; the expression of pitA is constitutive under different inorganic phosphate concentrations while that of pitH is induced in condition of low phosphate availability [58, 68].
Page 8, Figure 2 legend
- The gene sco4144located behind ppk encodes a phosin bearing a CHAD domain that is considered as a polyphosphate binding module ( Werten et al 2019). This enzyme is considered as an accessory factor involved in the upkeep of polyP in a conformation suitable for its use by Ppk and/or its degradation by exopolyphosphate phosphatases.Sco4143 encodes a protein of the nudix hydrolase family bearing similarities with enzymes involved in the cleavage of diadenosine hexaphosphate (Ap6A) into two molecules of ATP. Furthermore, it would be nice to provide SCO names for the genes mentioned.
Answer: We have added a new paragraph in lines 378 is indicated “In addition, the gene SCO4144, downstream of ppk, encodes a protein with 67% identity to the phosin PptA (Werten et al., 2019) that binds methals and polyphosphate (Werten et al., 2019). SCO4144 contains all the histidine (H) residues characteristics of metal-binding proteins and the arginine (R) residues for sulphate binding found in PptA. The bioinformatic analysis of the SCO4143 protein reveals high similarity to Streptomyces nudix hydrolase family proteins involved in dinucleotides hydrolysis and we designated it nudA”. Also the legend of Fig 2 (line 334) has been modified.
- What doespap (SCO4137) located upstream of pitH1 stand for?
Annotation suggest that it might be related to PhoU ?
Answer: The role of the ancillary protein pap is unknown, although this gene is very well conserved. The pap name means pit-ancillary-protein
Page9
1) “This finding is interesting since it indicates that Streptomyces species have integrated these genes in a short region of the genome [76]. Such organization may benefit to the assimilation of phosphate, its storage as polyphosphate and/or the degradation of the latter”. In fact nobody knows what drives polyphosphate biosynthesis in Streptomyces as in most other micro-organisms but PhoU seems to be involved ( DiCenzo et al, 2017; Morohoshi et al 2002).
Answer: This paragraph has been partially modified in line 386 according to the reviewer as follows: “Such organization may benefit the assimilation of phosphate, its storage as polyphosphate and/or the degradation of the latter [77, 78, DiCenzo, Morohoshi]”
Page 10
- “These results suggest that in contrast to coli, glycerophosphodiesters are not transportedas such in Streptomyces but are hydrolyzed by extracellular phosphodiester phosphodiesterases that scavenge the phosphate group.”
Answer: The sentence has been modified as suggested in line 443
Page 11
- “Independently of whether sugar phosphate transport exist, it is possible that these sugar phosphates trigger a signalingcascade in Streptomyces sp. by interacting with membrane receptors.” That is pure speculation…
Answer: This sentence has been removed
2)“In S. coelicolor addition of extracellular ATP have a dose-dependent effect up to 10 mM ATP, increasing production of actinorhodin by stimulating expression of actII-orf4 regulatory gene but it was repressive at higher concentration [92].”Millan Oropesa
This observation can perhaps be related to the fact that S. coelicolor has a 2 to 3 fold higher intracellular ATP concentration than S. lividans (Independently of whether sugar phosphate transport exist, it is possible that these sugar phosphates trigger a signaling cascade in Streptomyces sp. by interacting with membrane receptors.” High ATP concentration can signals active oxidative metabolism as well as perhaps also growth slow down (ATP generation exceeding ATP needs for anabolism)
Answer: The sentence has been modified in lines 521-527 as follows “In S. coelicolor addition of extracellular ATP have a dose-dependent effect up to 10 mM ATP, increasing production of actinorhodin by stimulating expression of actII-orf4 regulatory gene but it was repressive at higher concentration [92]. These authors did not study the transport of ATP and suggested that ATP might have a regulatory role by interacting with a specific receptor in the cell membrane. The differential effect of ATP on antibiotics formation in S. coelicolor may be also related to the high intracellular ATP concentration of Streptomyces specie (Millan Oropeza et al)
Page 12
“Mutants disrupted in the phn gene cluster were unable to degrade simple phosphonates [57, 95].” This sentence is unclear please re-formulate.
Answer: The sentence has been modified in line 550 as follows “Mutants disrupted in phn genes were unable to degrade simple phosphonates”
Page 14
That is unclear whether the presence / absence of phn genes have been correlated or not with the ability to degrade phosphonates?
Answer: Line 614 and following. As explained in the first version of the text the evidence available indicates that simple phosphonates are transported and utilized by some Streptomyces sp. (Obojska et al., 1999) although further biochemical studies are required to confirm which of the component enzymes of the C-P lyase complex are really needed for the phosphonate degradation
Page 15
- a) The sentence “An important observation is that three genes cluster involved in phosphate metabolism are associated in a short region of Streptomyces genomes forming a phosphate metabolism supercluster.” is repeated in line 649-641 and in lines 657-659.
Answer: The second duplicated sentence has been eliminated
- b) Do plants produce numerous C-P compounds?
Answer: It seems that C-P compounds are present among the metabolites of plants, and more important the available evidence indicates that C-Ps in plants derive from external herbicides used in the agriculture
The word signalling is spelled in United Kingdom (not in USA) with two L according to the wordreference dictionary. This spelling is maintained in the text
Reviewer 2 Report
Martin and Liras have submitted a comprehensive discussion of mechanisms for maintenance of phosphate homeostasis in specific bacterial lines. The authors are to be congratulated on the level of detail and the scope of this project. As an essential element, phosphate homeostasis is finely tuned in these organisms and the identified mechanisms for sensing, transport and signaling have yielded insights into mammalian phosphate homeostasis. Furthermore, discovering the phosphate metabolic properties of bacteria has the potential for harnessing them for environmental clean up as well as limiting their toxicity in humans. Thus the topic is of importance. The included figures are very helpful. This reviewer has several suggestions that would strengthen the work considerably.
1. The overall organizing theme of the work is not clear. The intention to discuss high and low affinity phosphate transport systems is stated but within these two major categories, the internal organization is unclear. If the theme is going to be contrasting the high and low affinity systems, then a few sentences to explain why two systems are needed and when one or the other predominates would be worthwhile.
2. The authors bring up some incredibly interesting and important topics such as the ability to distinguish between phosphate and arsenate, the different phosphorus containing substrates that can be transported, and phosphate homeostasis gene "clusters". However, in the present form, the reviewer feels that the important messages are being lost at times. While the authors sometimes introduce a topic with a justification for its discussion and sometimes sum up the topic at the end of the section, this is not always the case. Adding these explanatory sentences consistently would be very helpful.
3. The authors highlight differences in the phosphate homeostasis components between different bacterial strains. It would be useful to the reader for the authors to provide some context. For example, why would one species utilize a specific form of phosphate while another does not? is it location? availability of substrate? lifecycle considerations?
4. As in any field, there terms that are commonly used and understood by workers in the area but may not be familiar to all readers. The reviewer notes that most of the time the authors have defined their terms but not always. The review would significantly increase its accessibility to readers who are new to the field by defining all of the terms such as phosphate, phosphite, phosphonate, candicidan, and DRu among others.
5. The meaning of some passages is not clear.
Lines 412-418. why is this important?
Lines 428-430. neither the significance nor meaning is clear
Lines 440-448. the authors have introduced the idea of regulation by carbon sources but should indicate the significance
Lines 461-462. what is the significance of the fact that C glutamicum has a glycerol 3 phosphate but not a G6P transporter?
Lines 494-498. similar concern
Lines 542-544. It is not clear how this last sentence relates to the rest of the paragraph
Lines 571-585. What is the conclusion of these observations?
6. The paragraph encompassing lines 638-643 is very provocative but would be strengthened by adding evidence or justification for the final sentence.
7. the authors may want to consider a table or figure detailing the similarities and differences between the bacterial species that they have discussed with regard to mechanisms of sensing, utilization of different phosphate substrates, etc.
Minor issues with English
1. line 95 not, not no
2. line 107 "allowed to locate"? not sure what you mean
3. line 153 replete, not repleted
4. line 264 transports, not transport
5. line 265 does not, not do not
6. line 325 respond to, not responds to
7. line 366 have integrated, not integrate
8. line 367 benefit, not benefits
9. line 431 correspond, not corresponds
10. line 477 did not find, not found
11. line 494 extracellular ATP has, not have
12. line 514 phosphonate, not phosphonates. performed would be a better word than made
13. line 517, think the word "it" should precede "is" at the end of the line
14. line 533, have, not has
15. line 600, utilizes, not utilize. in THE presence, not in presence
Author Response
- The overall organizing theme of the work is not clear. The intention to discuss high and low affinity phosphate transport systems is stated but within these two major categories, the internal organization is unclear. If the theme is going to be contrasting the high and low affinity systems, then a few sentences to explain why two systems are needed and when one or the other predominates would be worthwhile.
Answer: The following sentences have been added in line 92: “The wide distribution of both the low affinity and high affinity systems in Gram positive and Gram negative bacteria indicates that there are physiological advantages in having two different transport systems. The low affinity Pit system requires less energy for phosphate transport since it is energized by a proton-antiporter system and is used in high phosphate concentration habitats. The high affinity PstSCAB requires more energy for phosphate transport since it is energized by ATP hydrolysis. This high affinity system is used in habitats or environmental conditions in which phosphate is limiting.
- The authors bring up some incredibly interesting and important topics such as the ability to distinguish between phosphate and arsenate, the different phosphorus containing substrates that can be transported, and phosphate homeostasis gene "clusters". However, in the present form, the reviewer feels that the important messages are being lost at times. While the authors sometimes introduce a topic with a justification for its discussion and sometimes sum up the topic at the end of the section, this is not always the case. Adding these explanatory sentences consistently would be very helpful.
Answer: The introduction to the major items in the article has been revised and new introductory sentences have been added in lines 231, 382 and 529 as follows:
- -Line 231 : Added “The PstS proteins are usually cell membrane-anchored proteins in different bacteria. However, surprisingly it was reported that in some Streptomyces species this protein was released to the culture medium”.
- -Line 374: Added “The location of the phosphate transport genes and particularly their possible organization in clusters is of high interest since it will contribute to understand their expression”
- -Line 529 Another phosphate source that might be utilized by microorganisms are phosphonates.
- The authors highlight differences in the phosphate homeostasis components between different bacterial strains. It would be useful to the reader for the authors to provide some context. For example, why would one species utilize a specific form of phosphate while another does not? is it location? availability of substrate? Lifecycle considerations?
Answer: This question was addressed in lines 28, 490-495 and 680 in the case of pathogenic Corynebacteria which have genes for utilization of sugar phosphates which are absent in saprofitic Corynebacteria. A new paragraph has been added in Conclusions (line 678) indicating “There are important differences in the ability of bacteria to transport and grow on distinct phosphate substrates; this is a normal behaviour since phosphate in nature is present in quite different forms in distinct habitats and the bacteria face the challenge to utilize the phosphate form that is available. In contrast to pathogenic bacteria most Streptomyces species are soil habitants and therefore have powerful systems to scavenge phosphate from soil including rock phosphate”
- As in any field, there terms that are commonly used and understood by workers in the area but may not be familiar to all readers. The reviewer notes that most of the time the authors have defined their terms but not always. The review would significantly increase its accessibility to readers who are new to the field by defining all of the terms such as phosphate, phosphite, phosphonate, candicidin, and DRu among others.
Answer: Additional definition have been added to clarify the points raised by the reviewer. In lane 199 we have added before the word candicidin “the polyene macrolide antibiotic” to indicate that this compound is an antibiotic.
-In line 127 we explained that Dru is a direct repeat unit
-The definition of phosphonate was already included in lines 29 and 530
-In line 285 we have added “The phosphites are salts of phosphorous acid”.
5.The meaning of some passages is not clear:
a) Lines 412-418. why is this important?
Answer: The transport of intact sugar phosphate is important because it avoid hydrolysis to separate the phosphate from the sugar. When the sugar phosphates are transported intact they may be utilized directly in the glycolysis and this saves energy.
Lines 428-430. neither the significance nor meaning is clear
Answer: We have modified the sentence removing the word cognate to made it clear. In line 494 now says “All these sugar phosphates, but not their dephosphorylated form, delayed the differentiation…”
c)Lines 440-448. the authors have introduced the idea of regulation by carbon sources but should indicate the significance
Answer: The regulation by carbon sources is not the aim of this study. We simply state that the glycerolphosphate phosphodiesterases were reported to be regulated by serine, and other carbon sources.
d)Lines 461-462. what is the significance of the fact that C glutamicum has a glycerol 3 phosphate but not a G6P transporter?
Answer: As indicated above (lines 490 and 680) pathogenic Corynebacteria may use G6P from the infected host tissues whereas Gly-3-P is usually commonly used by saprofitic Corynebacteria
e) Lines 494-498. similar concern
Answer: This was already answered in previous question
f) Lines 542-544. It is not clear how this last sentence relates to the rest of the paragraph
Answer: The sentence is scientifically correct but we have removed it from the text to facilitate understanding
g) Lines 571-585. What is the conclusion of these observations?
Answer: This suggests that some Arthrobacter species have acquired the phosphonate utilization genes by horizontal gene transfer from other soil bacteria and have retained them to grow on phosphonate rich habitats
- The paragraph encompassing lines 638-643 is very provocative but would be strengthened by adding evidence or justification for the final sentence.
Answer: We agree with the reviewer; this hypothesis is supported by the work of Gebhard et al (2006) that concluded that the phnCDE of M. smegmatis is now a phosphate transporter (not a phosphonate transporter). As we indicated in the text further research is required to confirm this point.
- the authors may want to consider a table or figure detailing the similarities and differences between the bacterial species that they have discussed with regard to mechanisms of sensing, utilization of different phosphate substrates, etc.
Answer: The differences in the transport system between bacteria are so large that it is very difficult at present time to stablished a comparative Table.
Minor issues with English
- line 95 not, not no. Corrected in line 102
- line 107 "allowed to locate"? Modified to “allowed to identify the pstSCAB operon” in line 114
- line 153 replete, not repleted. Modified in lines 165 and 170
- line 264 transports, not transport. Modified in line 283
- line 265 does not, not do not….247. Corrected in line 284
- line 325 respond to, not responds to…Corrected in line 335
- line 366 have integrated, not integrate. Corrected in line 385
- line 367 benefit, not benefits. Corrected in line 386
- line 431 correspond, not corresponds. Corrected in line 458
- line 477 did not find, not found. Corrected in line 504
- line 494 extracellular ATP has, not have. Corrected in line 521
- line 514 phosphonate, not phosphonates. performed would be a better word than made. Corrected in line 544
- line 517, think the word "it" should precede "is" at the end of the line, Corrected in 544.
- line 533, have, not has. We have changed the phrase to made it more clear in line 571
- line 600, utilizes, not utilize. in THE presence, not in presence. Corrected in line 628.
Round 2
Reviewer 2 Report
The reviewer thanks the authors for their responses.